# Conditions Leading to Elevated PM_2.5_ at Near-Road Monitoring Sites: Case Studies in Denver and Indianapolis

**DOI:** 10.3390/ijerph16091634

**Published:** 2019-05-10

**Authors:** Steven G. Brown, Bryan Penfold, Anondo Mukherjee, Karin Landsberg, Douglas S. Eisinger

**Affiliations:** 1Sonoma Technology, Inc., Petaluma, CA 94954, USA; bryan@sonomatech.com (B.P.); amukherjee@sonomatech.com (A.M.); doug@sonomatech.com (D.S.E.); 2Washington State Department of Transportation, Olympia, WA 98504, USA; LandsbK@wsdot.wa.gov

**Keywords:** near-road, PM_2.5_, Denver, Indianapolis

## Abstract

We examined two near-road monitoring sites where the daily PM_2.5_ readings were among the highest of any near-road monitoring location in the U.S. during 2014–2016: Denver, Colorado, in February 2014 and Indianapolis, Indiana, in November 2016. At the Denver site, which had the highest measured U.S. near-road 24-hr PM_2.5_ concentrations in 2014, concentrations exceeded the daily National Ambient Air Quality Standards (NAAQS) on three days during one week in 2014; the Indianapolis site had the second-highest number of daily exceedances of any near-road site in 2016 and the highest 3-year average PM_2.5_ of any near-road site during 2014–2016. Both sites had hourly pollutant, meteorological, and traffic data available, making them ideal for case studies. For both locations, we compared air pollution observations at the near-road site to observations at other sites in the urban area to calculate the near-road PM_2.5_ “increment” and evaluated the effects of changes in meteorology and traffic. The Denver near-road site consistently had the highest PM_2.5_ values in the Denver area, and was typically highest when winds were near-downwind, rather than directly downwind, to the freeway. Complex Denver site conditions (near-road buildings and roadway alignment) likely contributed to higher PM_2.5_ concentrations. The increment at Indianapolis was also highest under near-downwind, rather than directly downwind, conditions. At both sites, while the near-road site often had higher PM_2.5_ concentrations than nearby sites, there was no clear correlation between traffic conditions (vehicle speed, fleet mix) and the high PM_2.5_ concentrations.

## 1. Introduction

Air pollution is recognized to be elevated near major roads [1,2,3,4,5] and, in 2010, the U.S. Environmental Protection Agency (EPA) mandated that near-road air quality be measured in selected U.S. metropolitan areas [6]. As of 2018, 70 near-road monitors were operational under the EPA mandate, measuring a number of pollutants. Near-road air pollution is a mixture of gaseous and particulate species such as the criteria pollutants, NO_2_, CO, and PM_2.5_, as well as other trace species such as black carbon and air toxics. In this paper, we examine two cases—Denver, Colorado, and Indianapolis, Indiana—where some of the highest near-road PM_2.5_ concentrations have been observed under the U.S. near-road monitoring program. These in-depth case studies complement separately published national-scale reviews of EPA-mandated near-road air pollutant measurements [7,8]. Since there are EPA requirements to conduct PM_2.5_ hot-spot assessments for certain transportation highway projects that involve significant levels of diesel vehicle traffic or are otherwise designated as being of localized air quality concern [9], there is a need to understand the contribution from roadways on PM_2.5_ next to roadways. Data from air quality monitors deployed next to major U.S. roadways in response to the EPA mandate offers an opportunity to examine how near-road PM_2.5_ varies with meteorology and traffic over short- and long-term periods.

PM_2.5_ comprises a mixture of carbonaceous, inorganic, semi-volatile, and metal species. In the near-road environment, it includes direct emissions of PM and gaseous PM precursors from tailpipe exhaust that are predominantly carbonaceous, as well as emissions of metals from brake or tire wear and lubricating oil [10,11,12,13]. Gaseous precursors in exhaust can rapidly react and condense to form secondary organic particles within seconds or minutes [14,15,16,17,18,19,20,21,22,23]. These reactions and the formation of secondary PM_2.5_ depend not only on the concentrations of gaseous precursors but also on the concentrations and composition of existing particles, contributing to a complicated, dynamic environment of carbonaceous particles next to a roadway. Resuspension of road dust also contributes to PM_2.5_ and PM_10_ concentrations. Road dust can include typical soil elements such as iron and calcium, as well as trace metals such as zinc, and its composition can vary from location to location [10,24]. With this mixture of species, exposure to near-road pollution has been linked to a variety of health effects [1,25,26,27]. It is estimated that roughly 20% of the U.S. population lives next to roadways [28]; therefore, understanding the conditions that lead to high concentrations of PM_2.5_, including whether they coincide with other mobile source-related species such as CO or NO_2_, is of considerable interest.

Prior work focused on comparing CO, PM_2.5_ and NO_2_ concentrations among near-road sites throughout the country [7,8]. As part of that work, the average increment in PM_2.5_ between a given near-road site and nearby sites was assessed on an annual basis. Here, we present a day-by-day look at PM_2.5_ concentrations at two near-road sites in comparison to nearby sites, traffic, and meteorological conditions. The cases examined represent the highest measured U.S. near-road 24-hr PM_2.5_ concentrations in 2014 (Denver), and the highest measured U.S. near-road 3-yr annual average PM_2.5_ concentrations for 2014–2016 (Indianapolis).

Indianapolis and Denver have a different mix of sources and meteorology impacting local PM_2.5_ concentrations. Indianapolis lies at a confluence of major interstate freeways and is subject to a number of point sources within or near the city. Denver is on the eastern side of the Rocky Mountains, with a predominantly north-south wind pattern; when winds are from the west, wind speed is typically high. Sources of PM_2.5_ in Denver are predominantly mobile sources and small-scale point or area sources, and PM_2.5_ is typically a combination of carbonaceous particles and ammonium nitrate [29,30,31].

For each Denver and Indianapolis near-road site, we use a year of hourly near-road PM_2.5_ measurements paired with hourly measurements at a nearby, urban-scale monitoring site to examine whether PM_2.5_ concentrations next to these roadways are higher under direct downwind conditions or when the monitoring site is downwind at an angle across the roadway. Prior work has typically focused on annual average trends at multiple near-road sites or measurements at a single near-road site during a limited time of the year. This work combines short- and long-term data analysis to derive insights into roadway contributions to near-road PM_2.5_. Here, for each of Denver and Indianapolis, we complete a detailed examination of meteorology, air quality, and traffic data during multi-day PM_2.5_ episodes and we analyze a year’s worth of hourly near-road data. To our knowledge, this is the first study that systematically identifies high-concentration near-road PM_2.5_ episodes from across the United States near-road monitoring network, evaluates the relationship between traffic conditions and episodic PM_2.5_, and contrasts how short-term (episodic) and long-term (annual) near-road PM_2.5_ concentrations relate to roadway traffic and regional background concentrations. Importantly, with a year’s worth of data to support statistical assessments, this work examines the variability of near-road PM_2.5_ increments under downwind compared to near-downwind conditions, and complements existing modeling and measurement literature. Given the U.S. EPA-mandated PM_2.5_ hot-spot analyses for certain transportation projects [9], this work takes advantage of data becoming available from the EPA-mandated near-road monitoring network to improve understanding of real-world relationships between traffic conditions and near-road PM_2.5_.

## 2. Methods

This case study analyses the conditions under which PM_2.5_ concentrations are higher at the near-road site, i.e., how changes in traffic or meteorological patterns impact PM_2.5_ concentrations next to the roadway. We examine data at the Denver Interstate Highway 25 (I-25) near-road site on three days during February 2014 when PM_2.5_ concentrations were above the NAAQS threshold of 35 µg/m^3^, and at the Indianapolis I-70 near-road site on three days during November 2016 when PM_2.5_ concentrations were above the NAAQS threshold of 35 µg/m^3^. Other near-road sites also exceeded this threshold on some days, but both the Denver and Indianapolis sites recorded hourly PM_2.5_, meteorological, and other pollutant data, plus hourly traffic data, making these sites ideal for case study analysis. In addition, when compared to other nearby sites, these two near-road sites have different estimated PM_2.5_ increments [8]. The Indianapolis site has a smaller estimated PM_2.5_ increment (0–1 μg/m^3^ across three methods) than the Denver near-road site (approximately 3 μg/m^3^ across three methods); as a result, these sites can provide a contrast for examining the conditions under which PM_2.5_ concentrations are elevated at a near-road site compared to nearby urban sites.

Hourly meteorological PM_2.5_ and NO/NO_2_/NO_x_ data were downloaded from the EPA’s Air Quality System (AQS) for all air monitoring sites, including the near-road site, in the Denver and Indianapolis areas for 2014 and 2016, respectively. The near-road site (AQS ID 08-031-0027) in Denver is 9 m from the east side of I-25, with an annual average daily traffic (AADT) count of 249,000 and a fleet-equivalent AADT (FE-AADT, a measure that weights trucks more heavily for count purposes) of 263,118 (see the EPA’s list of near-road sites at https://www3.epa.gov/ttnamti1/nearroad.html). Data from the near-road site were compared to data from nearby monitors, which include CAMP (AQS ID 08-031-0002, the monitor used to represent background in our analyses based on factors such as prevailing winds and site characteristics) 3 km to the northeast, 14th and Albion (AQS ID 08-031-0013) 6.5 km to the east, La Casa (AQS ID 08-031-0026) 5 km to the north and Alsup (AQS ID 08-001-0006) 12 km to the northeast. All sites are within the urbanized Denver area and can provide context for the near-road measurements. The PM_2.5_ was measured hourly via a GRIMM EDM 180 Federal Equivalent Method (FEM) at both the near-road and nearby CAMP site. The GRIMM at the near-road site has correlation (R) with the collocated 24-h FRM measurement of 0.95 and slope of 0.91, for the 2015–2017 (see https://www.epa.gov/outdoor-air-quality-data/pm25-continuous-monitor-comparability-assessments).

The Indianapolis near-road site (AQS ID 18-097-0087) is 24.5 m south of the I-70 freeway, with an AADT of 189,760 and an FE-AADT of 362,110. Nearby monitors used for comparison to the near-road site include Washington Park (AQS ID 18-097-0078), 3 km to the northeast, which is the monitor used to represent background in our analyses based on factors such as prevailing winds and site characteristics and W. 18th St. (AQS ID 18-097-0081) 7.1 km to the west. These monitors have hourly PM_2.5_, are within the urbanized Indianapolis area and are listed by the state monitoring agency as neighborhood- or middle-scale. The near-road site has hourly data via a Thermo Scientific 5030 SHARP instrument, and the nearby Washington Park site has hourly data via a MetOne BAM 1020, both of which are FEM. The daily averaged data at the near-road monitoring site has a correlation (R) of 0.93 and slope of 0.97 with the collocated FRM for 2015–2017, and the daily averaged data from the BAM at the Washington Park site has correlation (R) of 0.92 and slope of 0.89 with the collocated FRM.

The traffic monitor closest to the Denver near-road air monitoring site is located at I-25/6th Avenue, approximately 700 m south of the near-road monitoring site. Traffic data from this monitor were obtained via email contact with the Colorado Department of Transportation. Traffic data were the total number of vehicles and hourly measurements of northbound and southbound vehicle speed across ten bins. Hourly average vehicle speed was estimated by multiplying the number of vehicles in each speed bin by the midpoint of the bin, then summing these values and dividing by the total number of vehicles in the hour. Traffic data for Indianapolis were downloaded from the Indiana Department of Transportation’s traffic data repository (indot.ms2soft.com). Data were obtained by traffic direction (eastbound, westbound) at the location of the near-road air quality monitor. Data were hourly values in FHWA-Scheme F classification; i.e., they included a count of motorcycles, cars, light trucks, buses, and heavy-duty trucks, as well as the number of vehicles in each of 14 speed bins. Data were aggregated to the hourly average speed of all vehicles and hourly counts of total vehicles and of heavy-duty trucks, defined as those with three or more axles (FHWA Classes 6 and above).

We examined how the contribution of emissions from the roadway vary by wind direction. At both sites, hourly PM_2.5_ and meteorological data are available, with a nearby site available to support comparisons to background. Modeling of roadway emissions suggests that, especially under stable meteorological conditions, maximum concentrations from roadway emissions do not occur under wind directions when a receptor is perpendicular to the roadway, but rather at angles when the wind direction is not perpendicular to the roadway [32,33,34]. The rationale is that at non-perpendicular angles an air parcel travels over more roadway length, and receives more emissions, than a parcel that traveled perpendicular to the roadway. For example, a near-road study in Raleigh found that high concentrations of black carbon (BC) and other pollutants occurred under a wide array of wind directions, not just when the monitoring site was directly downwind [32]. Also, Venkatram et al. (2013) developed a line source model that compared well to data from published near-roadway tracer studies [35], and found that modeled concentrations were higher when winds were near-parallel to the roadway [32,33,34]. In their modeling work, they found that concentrations at 10 m downwind can reach up to 2.5 times the value seen normal (perpendicular) to the roadway, under stable meteorological conditions. Barzyk et al. (2009) found that an “effective distance” parameter, which incorporates the predominant wind direction and distance of a receptor to the roadway together, was a statistically significant predictor of mobile source air toxics, while exact distance to road was not as predictive [36].

Wind speed and direction are not the only meteorological factors that can impact near-road PM_2.5_, since relative humidity and atmospheric pressure can also modify the rapidly changing chemistry of particulate matter and its precursors. For example, in an urban street canyon, Richmond-Bryant et al. found that temperature was a significant predictor of near-road PM_2.5_ and black carbon (BC), in addition to background concentrations (as the strongest predictor), wind speed, wind direction and traffic [37]. Pearce et al. also found that temperature, water vapor pressure and boundary layer height, in addition to wind speed and direction, was predictive of PM_10_ in a generalized additive model (GAM) [38]. At a near-road site in Shanghai, Wang et al. found that, in addition to background concentration levels, air pressure, solar radiation and dew-point temperature were significant predictors of PM_2.5_ in a GAM [39]. Together, these and other observational and modeling studies show that the near-road environment is complex, with a myriad of meteorological influences on the evolution and ultimately the concentration levels of PM_2.5_ next to a roadway [15,18,19,20,40,41,42,43,44,45,46,47].

At both sites, we examined the hourly near-road PM_2.5_ increment, meaning the PM_2.5_ concentration difference between measurements taken next to the road compared to those from a nearby site used to represent background (CAMP for Denver, Washington Park for Indianapolis). The angle that the monitoring site was perpendicular to, and therefore at the minimum distance from, the adjacent roadway was determined in ArcGIS. At Denver, the perpendicular downwind compass point was at 255 degrees and at Indianapolis, it was 345 degrees. We defined the near-road monitor as being “downwind” of the road when winds flowed from the roadway toward the monitor, for all wind directions within a ± 15-degree arc of perpendicular to the roadway (i.e., a 30-degree arc in total). Similarly, we defined a “near downwind” bin for wind directions that were a further 30 degrees away from the downwind bin. We also established “near parallel” and “parallel” bins by adding additional 30-degree arcs, with the remaining spatial area defined as the “upwind” bin. Data were limited to only those hours with stable conditions (defined here as wind speed of less than 1 m/s) and hours when traffic was typically high (06:00 to 20:00). We then examined median hourly PM_2.5_ increment by wind direction bin, and used the Kruskal–Wallis test to determine whether increment concentrations at non-downwind angles were significantly different than concentrations at downwind angles. Figure 1 shows the case study sites and wind direction bins used at each site.

## 3. Results

### 3.1. Denver

Between 2 and 12 February 2014, four periods of relatively high PM_2.5_ concentrations were measured at the Denver near-road site. Hourly concentrations were high overnight on 3–4 February, during the morning and afternoon of 4 February, at midday on 7 February, and throughout 9 and 10 February. The 24-hr PM_2.5_ concentrations exceeded 35 μg/m^3^ on three days: 7 February (35.4 µg/m^3^), 9 February (44.4 µg/m^3^), and 10 February (57.0 μg/m^3^). The 10 February PM_2.5_ concentration was the highest 24-hr PM_2.5_ value measured in 2014 across all of the near-road monitoring sites in the U.S.

Figure 2 shows traffic volume and speed near the near-road site, hourly and 24-hr PM_2.5_, hourly NO/NO_2_/NO_x_, wind speed, and whether the near-road monitor was upwind or downwind of the freeway. Hourly PM_2.5_ concentrations do not exhibit a consistent diurnal cycle during this period (2−12 February 2014). In some cases (e.g., 2 and 7 February), hourly PM_2.5_ concentrations peaked in the middle of the day, while in other cases (e.g., 3 February), PM_2.5_ concentrations were highest during the overnight hours. Between 9 and 11 February, PM_2.5_ concentrations were consistently elevated for multiple consecutive days.

In general, PM_2.5_ concentrations were higher when wind speeds were lower. Although wind speeds vary from day to day, they are generally higher during midday and afternoon. When hourly PM_2.5_ concentrations were elevated, wind speeds were typically low, and winds were often from the north (i.e., roughly parallel to the freeway), varying between upwind and downwind conditions.

In some cases (e.g., 7 February), high hourly PM_2.5_ concentrations coincided with high hourly NO_x_ concentrations; however, during other periods of high PM_2.5_ concentrations (e.g., 4 February), NO_x_ concentrations were not elevated. NO_2_ concentrations were consistently low at the near-road site between 2 and 12 February. There is no consistent relationship between NO/NO_2_/NO_x_ or temperature and high PM_2.5_; when the temperature was below 20 °F from 5–6 February, PM_2.5_ was relatively low. PM_2.5_ had little to modest correlation with NO_2_ or NO_x_ (r^2^ = 0.26–0.28).

Lastly, traffic volumes exhibit a typical diurnal activity pattern, with morning and afternoon peaks consistent with the morning and evening commute times. Traffic speeds are somewhat variable and do not track the diurnal signature of traffic volumes. Neither NO_x_ nor PM_2.5_ concentrations were correlated with vehicle speeds or traffic volumes during 2−12 February 2014 (r^2^ values of less than 0.10).

The Denver site has additional characteristics that likely influence the measured near-road concentrations. Three factors in particular are worth noting: the roadway alignment, the presence of buildings next to the monitor, and the location of an onramp. The analysis here characterizes wind conditions that result in the near-road monitor being upwind or downwind of the road; these conditions are defined by the roadway alignment adjacent to the monitor. However, Figure 1 illustrates that I-25 does not continue in a straight line north and south of the monitor; the curvature of the road to the north and south results in a weak “C” shape visible in satellite imagery. When winds are roughly parallel to the road, sections of the road to the north and south may effectively be upwind of the monitor, albeit only slightly. In addition, the monitor is sited between I-25 and the rear side of buildings that create a nearly continuous wall facing I-25. These buildings may act as a solid barrier that prevents pollutant dispersion, leading to higher near-road pollutant concentrations than if the buildings were not present. Finally, there is an onramp southwest of the monitor that, as vehicles accelerate to enter I-25, may influence near-road PM_2.5_ concentration values, especially when winds are out of the southwest.

Next, we examined PM_2.5_ data from nearby sites to assess whether all sites varied together, indicating an urban-scale PM_2.5_ signature. Figure 3 shows a pollution rise in PM_2.5_ in February 2014 at the near-road site, as well as a time series of PM_2.5_ concentrations at the near-road site and at three nearby regional PM_2.5_ monitoring sites. The figure also shows the difference between the concentrations at the near-road site and the nearest site (CAMP) for February 2014. When winds are from the west, directly from the adjacent freeway, concentrations are typically lower than they are when winds come from other directions, particularly from north or south. The time series shows that hourly PM_2.5_ concentrations at the near-road site are closely correlated with concentrations at the nearby site (r^2^ of 0.96 between the near-road site and the background site, CAMP). PM_2.5_ concentrations are typically higher at the near-road site than at the regional sites. Hourly PM_2.5_ concentrations were higher at the near-road site than at the nearby regional CAMP, 14th and Albion, and La Casa PM_2.5_ monitoring sites during 71% of the hours in February 2014; 24-hr concentrations were higher at the near-road site than at the nearby regional sites on 79% of the days in all of 2014. Furthermore, concentrations at the near-road site were typically higher than concentrations at other sites when PM_2.5_ concentrations were high. For example, on 9 and 10 February, hourly PM_2.5_ concentrations at the near-road site were 7−12 μg/m^3^ higher than at other nearby sites for multiple hours. On all but one day of February 2014, 24-hr concentrations at the near-road site were the highest of any of the sites in Denver. As seen in the pollution rise, concentrations at the near-road site were consistently higher when winds came from the north or south.

We also examined whether the excess concentrations typically found at the near-road site occurred only when the near-road site was downwind of the freeway, and how they varied by wind direction. To better depict the variation of PM_2.5_ with wind direction during February 2014, Figure 4 shows wind direction versus the difference between the hourly PM_2.5_ concentrations at the near-road site and the nearby CAMP site. Observations are sized by wind speed and colored by PM concentration; note that wind direction measurements are highly uncertain under low-wind-speed conditions. The near-road site typically recorded the highest hourly concentrations of any site in Denver during February 2014. The roadway alignment at the monitor location may mean that the monitor is downwind from the roadway across roughly 180 degrees of wind directions, and adjacent buildings likely slow or prevent dispersion.

Since the highest PM_2.5_ concentrations in the region were consistently measured at the near-road site during February, we examined whether this was true throughout the year. Figure 5 shows the daily PM_2.5_ concentrations during 2014 for the Denver near-road site, the concentrations at the nearby CAMP site, and the difference in concentration between the near-road site and the CAMP site. PM_2.5_ concentrations were higher at the near-road site than at the nearby site on 91% of days in 2014, and higher than any other site in Denver on 79% of days.

We further examined how the difference in concentrations at the near-road site and the CAMP site (the PM_2.5_ increment) varied by hour and wind direction (Figure 6 and Table 1); differences were examined statistically based on a Kruskal–Wallis test. The PM_2.5_ increment is higher from the southwest near-downwind bin than from the downwind bin and northwest near-downwind bin, at a 95% confidence level. Increment differences were more pronounced when all hours were used, with a PM_2.5_ increment of 2.9 μg/m^3^ with near-downwind winds versus 1.7 μg/m^3^ with downwind winds. The distance an air parcel would travel over the roadway is longer for the near-downwind bins compared to the downwind bin (59 m compared to 50 m), though with near-downwind winds from the northwest, the difference to downwind concentrations is not significant. In addition to the increased travel distance over the roadway, emissions from vehicles accelerating on the on-ramp to the southwest may also be part of the increment in this southwest direction and may be the difference between the southwest and northwest near-downwind concentrations. Even though we have filtered data for low wind speeds, it is likely that other differences in stability not captured in the wind speed have an effect, causing the concentrations in the northwest near-downwind bin to be lower than those in the southwest near-downwind bin. For example, median wind speed is higher from the northwest near-downwind bin than from the southwest near-downwind bin (1.07 and 1.21 m/s, respectively). The increments are consistently higher for these bins during lower wind speed conditions. The highest increments are seen for these bins when restricting for wind speed less than 1 m/s. Overall, we do see a higher near-road increment of PM_2.5_ when the monitor is not directly downwind of the roadway but located at an angle near to downwind of the roadway.

In summary, the Denver near-road site consistently has the highest PM_2.5_ in the Denver area. As seen in specific high-PM_2.5_ events during February 2014 and year-round in 2014, PM_2.5_ concentrations at the near-road site are the highest of any monitor in the region on nearly 80% of days. The high near-road concentration events during February 2014 do not appear to be caused by unusual traffic conditions next to the monitoring site; the high concentrations are largely driven by regional PM_2.5_ conditions, yet the near-road site consistently has the highest concentrations in the region. On average for 2014, the near-road site is 1.5 μg/m^3^ and 16% higher than the maximum concentration of nearby sites. The near-road increment is highest with winds near-downwind from the southwest, i.e., when the monitor is not directly downwind of the freeway, though not when winds are from the northwest. Potential factors contributing to this outcome include higher wind speeds when winds are from the northwest compared to the southeast, the presence of buildings immediately east of the monitor that may trap roadway pollutants when winds are from the southwest, and vehicles accelerating onto the on-ramp to the southwest.

### 3.2. Indianapolis

The Indianapolis near-road site had the highest 3-year annual average (11.6 μg/m^3^) of any U.S. near-road site where three years of complete data through 2016 were available. In November 2016 at this site, PM_2.5_ was above 35 μg/m^3^ on three days, which are the focus of this case study: 5 November (38.5 µg/m^3^), 6 November (39.1 µg/m^3^), and 13 November (35.2 μg/m^3^). These days were all on weekends.

On weekdays, the traffic along I-70 peaked twice throughout the day, from approximately 06:00 09:00 LST (morning peak), and from 15:00 to 18:00 LST (evening peak). This pattern is typical of major freeways in urban areas. The highest PM_2.5_ concentrations occurred during the evening and early morning hours (approximately 18:00 to 08:00 LST), when the wind directions were from the east and southeast, though winds from these directions were relatively infrequent. The orientation of the I-70 with respect to the Indianapolis near-road site means that when winds are from the northwest, the site is downwind of the freeway. When winds were from the north/northwest, i.e., from the direction of I-70, PM_2.5_ concentrations were typically lower than when winds were from other directions.

Between 3 and 22 November 2016, several periods of relatively high PM_2.5_ concentrations were measured at the near-road site (Figure 7). High hourly concentrations consistently occurred in the late evening, overnight, and in the early morning. There was not a constant diurnal cycle of 1-h PM_2.5_ concentrations during 3–18 November 2016. On most days, PM_2.5_ concentrations were highest during the night. In general, PM_2.5_ was higher when wind speeds were slower. Although wind speeds varied from day to day, they were generally faster during midday and afternoon periods. When hourly PM_2.5_ concentrations were elevated, wind speeds were typically slow and the near-road monitor was more often upwind of I-70; this is true for 5, 6, and 13 November, when 24-h PM_2.5_ concentrations exceeded 35 μg/m^3^.

One indicator to help determine whether high levels of PM_2.5_ are associated with emissions from mobile sources is to examine whether NO_2_ concentrations are also elevated. During the period of 3–18 November, 1-h PM_2.5_ concentrations coincided with high hourly NO_x_ and NO_2_ concentrations (e.g., overnight peak on 5 November) at some times, but not at others (e.g., overnight peak 13 November). Traffic volumes exhibited a typical diurnal weekday activity pattern, with morning and afternoon peaks consistent with the morning and evening commute times. Traffic volumes on weekend days peaked near midday. NO_x_ concentrations and PM_2.5_ concentrations were poorly correlated with traffic volumes during these periods (r^2^ less than 0.10), and modestly correlated to each other (r^2^ = 0.47). Days when PM_2.5_ exceeded 35 μg/m^3^ (5, 6, and 13 November) were all weekend days, and while traffic measured on 5 November (Saturday) was higher than the mean traffic volume from October–December, the near-road monitoring site was predominantly upwind of I-70 on these days.

Given the ambient air quality data and traffic volumes, the elevated period of PM_2.5_ concentrations at the Indianapolis-0087 near-road monitoring site during November 2016 was likely not caused by on-road mobile emission sources. Rather, the diurnal pattern of 1-h PM_2.5_ may be indicative of the influence of residential wood burning (which generally causes elevated PM_2.5_ concentration at night when temperatures are low), overnight temperature inversions in the lower boundary layer (which exacerbate air quality conditions overnight and in the early morning, especially when wind speeds are slow), or both. To investigate this further, PM_2.5_ concentrations from the near-road monitoring site were compared to those from the nearby air quality monitoring site. Homogeneous PM_2.5_ concentrations would indicate a regional-scale influence on air quality, rather than impacts from local emission sources at the near-road site. During 3–18 November, 24-h PM_2.5_ concentrations at the near-road site were closely correlated with those at nearby sites (r^2^ of 0.96 between the near-road site and nearby Washington Park 0078, the background site) (Figure 8). PM_2.5_ concentrations were typically higher at the near-road site than at the regional sites during this period, by up to 9 μg/m^3^. Furthermore, 24-h PM_2.5_ concentrations were typically higher at the near-road site than at other sites when the absolute magnitude of PM_2.5_ concentrations was highest.

While 24-h PM_2.5_ concentrations were higher at the Indianapolis near-road site than at the background site during 3–18 November 2016, this was not always the case in 2016. The 24-h PM_2.5_ concentrations measured at the near-road site were higher than the nearby Washington Park site on 68% of days in 2016. Table 2 and Figure 9 show hourly PM_2.5_ increments at the near-road site during 2016, grouped by wind direction bins. Similar to results at Denver, the highest increment is seen when winds are near-downwind, in this case from the northerly near-downwind bin. Again, similar to Denver, there is asymmetry in the results by wind direction bin, where the increment is higher on the northern near-downwind side of the downwind bin but not on the western near-downwind side. Wind speed is also slightly lower in the northern near-downwind bin (1.1 m/s, median) compared to western near-downwind bin (1.5 m/s, median), which could also lead to the higher concentrations. Increments are consistently higher for these near-downwind bins during lower wind speed conditions, where the largest increments occur with winds less than 1 m/s; increments in both bins are significantly higher than those in the downwind bin under such conditions.

Overall, the average estimated near-road increment at the Indianapolis near-road site during all of 2016 was 1.0 μg/m^3^ when comparing to the nearby site; during 3–18 November, the near-road site typically had the highest concentrations of any site in Indianapolis, with an average increment of 4.5 μg/m^3^ when compared to the background site. On average in 2016, the near-road site is 1.1 μg/m^3^ and 13% higher than the Washington Park site.

## 4. Discussion

These two case studies exhibit some of the highest U.S. near-road PM_2.5_ concentrations observed by the EPA-mandated near-road air quality monitoring network in 2014–2016. In both cases, PM_2.5_ concentrations at the near-road monitoring site were above NAAQS thresholds, and were higher than at nearby sites. However, in neither case was there a clear indicator in traffic data (congestion, increased number of vehicles or trucks) that would suggest that PM_2.5_ should be higher at the near-road site than at other nearby sites. The near-road site was not downwind of the freeway more often than usual during these periods; an exception may be Denver, depending on the extent to which roadway alignment effectively broadened the wind conditions when the monitor was downwind of the road. Lastly, NO_2_, NO, and NO_x_ varied widely during these periods and did not correspond to high PM_2.5_ (r^2^ values less than 0.30). Rather, PM_2.5_ was consistently higher next to the roadway, with significant variations in its near-road increment depending on meteorology.

No speciation data are available at these two near-road sites to further examine the PM_2.5_ composition and the specific elements or particulate species that led to the PM_2.5_ near-road increment. Jeong et al. reported similar results in Toronto, where a near-road site had higher concentrations than a nearby site; by apportioning the components of PM_2.5_ through high-time-resolution measurements of PM_2.5_ species, they found that traffic-related sources of PM_2.5_ were two to three times higher next to the near-road site than at a nearby site [12]. The traffic-related sources included exhaust emissions plus brake/tire wear and re-suspended road dust, and the non-exhaust sources did correlate with the number of heavy-duty vehicles, though the total mass was small (roughly 5–10% of PM_2.5_). Results from Jeong et al. and our results are similar to those of other studies with a roadway/background site pair, where some organic compounds and black carbon are higher next to the roadway compared to a nearby or background site [13,16,48,49]. This combination may lead to significantly higher near-road concentrations of trace metals or other species and a modest increase in total PM_2.5_ mass. Over the course of 24 h, and relying on total PM_2.5_ mass as we have in our case studies here, any direct link between changes in traffic and the near-road increment of PM_2.5_ mass may be masked.

This is further seen in how the hourly increment varies by wind direction. At the Denver and Indianapolis sites, the PM_2.5_ increment is higher not when winds are directly perpendicular to the roadway, but at a slightly larger angle: near-downwind. Again, at both sites, there is asymmetry in the results, likely due to differences in wind speed and site characteristics such as the on-ramp next to the Denver site. These results support the concept that the combination of wind direction and distance is an important predictor of near-road concentrations, as seen in Bryzk et al., in modeling work by Venkatram et al., and elsewhere. In addition, examining hourly variations in other pollutants that have a larger near-road gradient than PM_2.5_, such as black carbon or ultrafine particles, could find further evidence of how concentrations may be higher at angles other than directly downwind. As additional, high frequency data are collected at multiple near-road sites over multiple years, future work can build on analyses presented here to further determine when emissions on a roadway impact nearby receptors and the implications for health impacts as well as meeting air quality standards.

## 5. Conclusions

We combined the analysis of short-term PM_2.5_ episodes at two near-road monitoring sites with high frequency data collected for an entire year at both sites to understand under what conditions near-road PM_2.5_ is high. Near-road PM_2.5_ was consistently higher at two near-road locations in Indianapolis and in Denver than at nearby locations. No direct link between changes in traffic was found to explain the day-by-day PM_2.5_ increment between the near-road and nearby sites, and the increment was highest on an hourly basis when winds were near-perpendicular to the roadway, rather than directly perpendicular. The relatively small but consistent difference in near-road and nearby PM_2.5_ concentrations is consistent with literature finding that road dust, break/tire wear, and black carbon are the main components resulting in higher PM_2.5_ concentrations next to the roadway, and that near-road PM_2.5_ concentrations averaged over the long-term are dominated by background concentrations. Future work could further examine near-road-speciated PM_2.5_ concentration data, a broader array of sites with varying measured near-road increments, and the differences between measured and modeled concentration data in the near-road setting.

## Figures and Tables

**Figure 1 ijerph-16-01634-f001:**
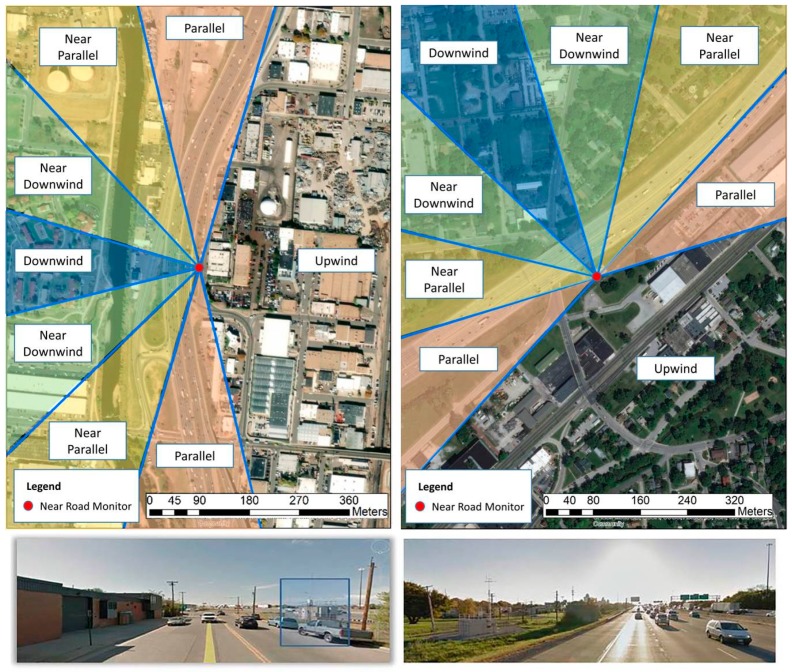
Satellite (**top**) and ground-level (**bottom**) views via Google Earth of the near-road sites at Denver (**left**) and Indianapolis (**right**); the Denver monitoring site is somewhat obscured by cars in the image, so the blue box denotes its location. In both satellite views, the angles used to determine wind direction bins (downwind, near downwind, near parallel, parallel and upwind) are also shown.

**Figure 2 ijerph-16-01634-f002:**
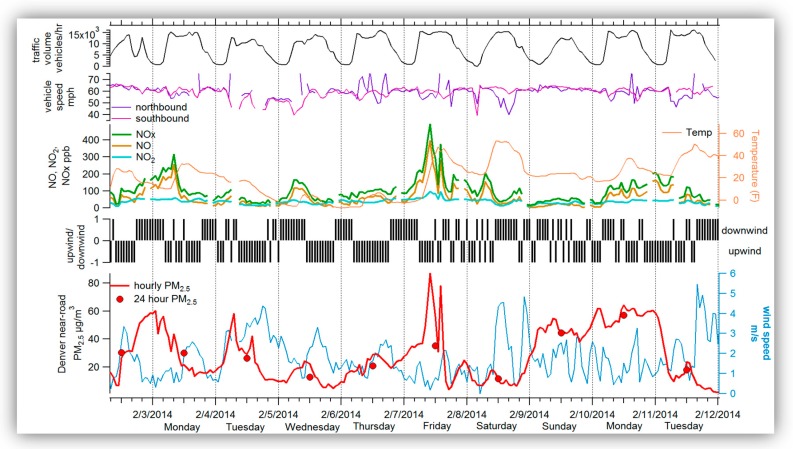
Denver case characterization. Hourly and 24-hr PM_2.5_ concentrations (μg/m^3^, red line), 24-hr PM_2.5_ concentrations (μg/m^3^, red dots), wind speed (m/s, blue line), whether the monitoring site was upwind (0 to 180 degrees) or downwind (180 to 360 degrees) of the freeway (black bars), NO_x_ concentrations (ppb, green line), NO concentrations (ppb, gold line), NO_2_ concentrations (ppb, teal line), and temperature (°F, orange line) at the near-road site in Denver during 2−12 February 2014. Also shown are vehicle speeds on I-25 (northbound, purple line; southbound, dark pink line) and traffic volume on I-25 (black line). Traffic data are for I-25 at 6th Ave., approximately 700 m south of the monitoring site.

**Figure 3 ijerph-16-01634-f003:**
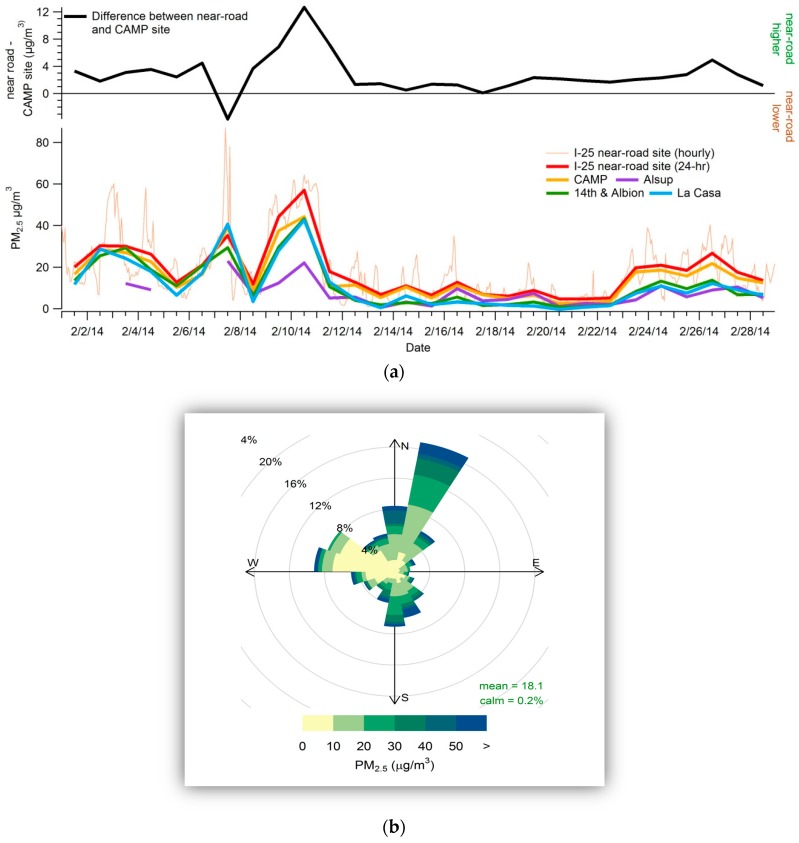
Denver near-road PM_2.5_ increment and pollution rose. (**a**) Hourly (thin orange line) and 24-hr PM_2.5_ concentrations (μg/m^3^, thick colored lines) at the near-road and nearby sites during February 2014, and the difference between 24-hr PM_2.5_ concentrations (μg/m^3^) at the near-road site and the nearby CAMP monitoring site (black line). (**b**) Pollution rose for hourly PM_2.5_ concentrations (µg/m^3^) at the Denver near-road site in February 2014. The size of each wedge indicates the frequency of wind direction. (For example, winds were out of the northeast nearly 25% of the time.) Color bands indicate the relative fraction of the time that concentrations occurred for each wind direction. For example, when winds were from the northeast, concentrations of 10–20 µg/m^3^ were most frequent, followed by concentrations of 20–30 µg/m^3^.

**Figure 4 ijerph-16-01634-f004:**
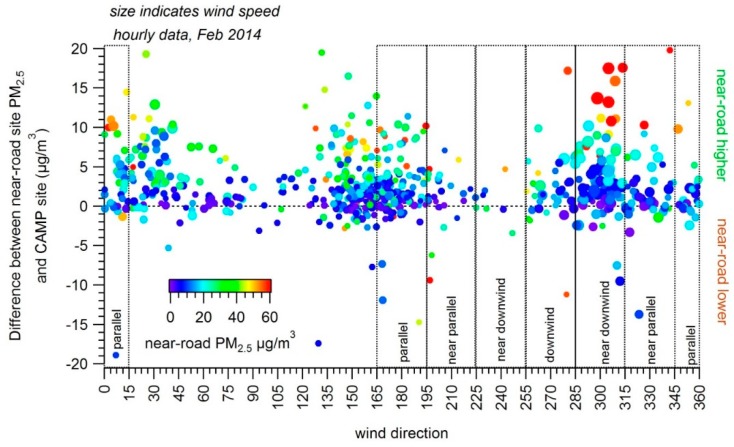
Denver PM_2.5_ increment vs. winds. Wind direction is shown versus the difference between hourly PM_2.5_ concentrations at the near-road site and the hourly PM_2.5_ concentrations at the CAMP site for data collected during February 2014. Larger points indicate higher wind speed, and colors indicate near-road PM_2.5_ concentration. Points above the horizontal dotted line indicate that hourly PM_2.5_ concentrations at the near-road site were higher than concentrations at the CAMP site, whereas points below the line indicate that hourly PM_2.5_ concentrations at the near-road site were lower than the CAMP site. Wind direction bins as defined in Figure 1 are also shown.

**Figure 5 ijerph-16-01634-f005:**
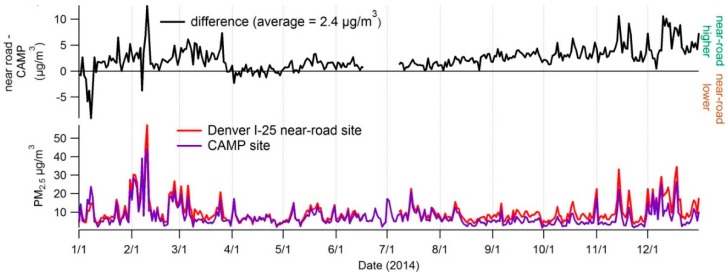
Daily PM_2.5_ concentration at the official Denver near-road site, the maximum concentration of all other nearby sites, and the difference between the two concentrations during 2014.

**Figure 6 ijerph-16-01634-f006:**
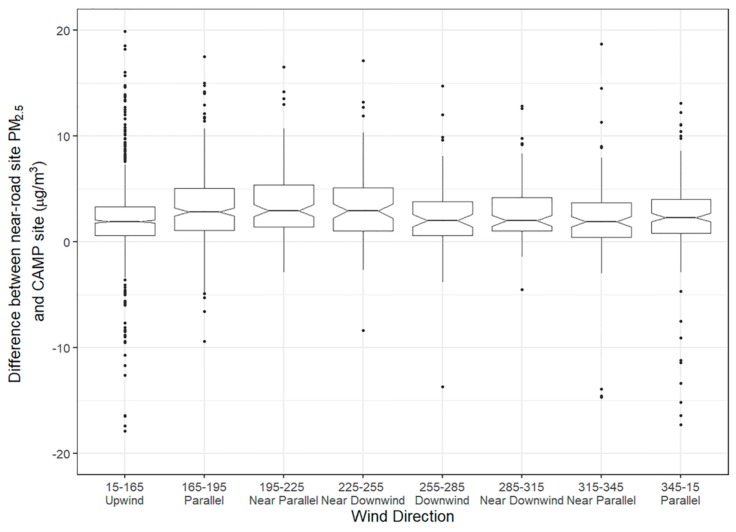
Notched box plot of the Denver hourly PM_2.5_ increment (difference between near-road and CAMP site) by wind direction bin during 06:00 to 20:00 and when winds speeds are below 2 m/s. The notch shows the 95% confidence interval on the median and the boxes show the interquartile range (IQR); points beyond the IQR are within the notch or are plotted individually.

**Figure 7 ijerph-16-01634-f007:**
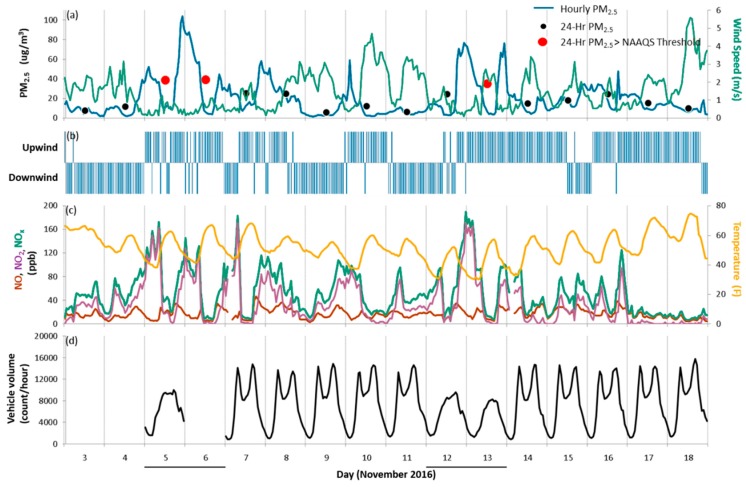
Indianapolis case characterization. Time series of meteorological, air quality, and traffic-related data from 3–18 November 2016, at the Indianapolis-0087 near-road monitoring site: (**a**) 1-h PM_2.5_ concentrations (blue line), 24-hr PM_2.5_ concentrations (black and red dots, where red dots highlight 24-h PM_2.5_ concentrations that exceeded the 35 μg/m^3^ NAAQS threshold), and wind speed (green line); (**b**) wind component (downwind means the near-road monitor was downwind of I-70, with winds originating from 240 to 60°); (**c**) NO (red line), NO_2_ (purple line), and NO_x_ (green line) concentrations, and temperature (orange line); and (**d**) traffic volume (vehicle count per hour). Weekend days are underlined along the x-axis.

**Figure 8 ijerph-16-01634-f008:**
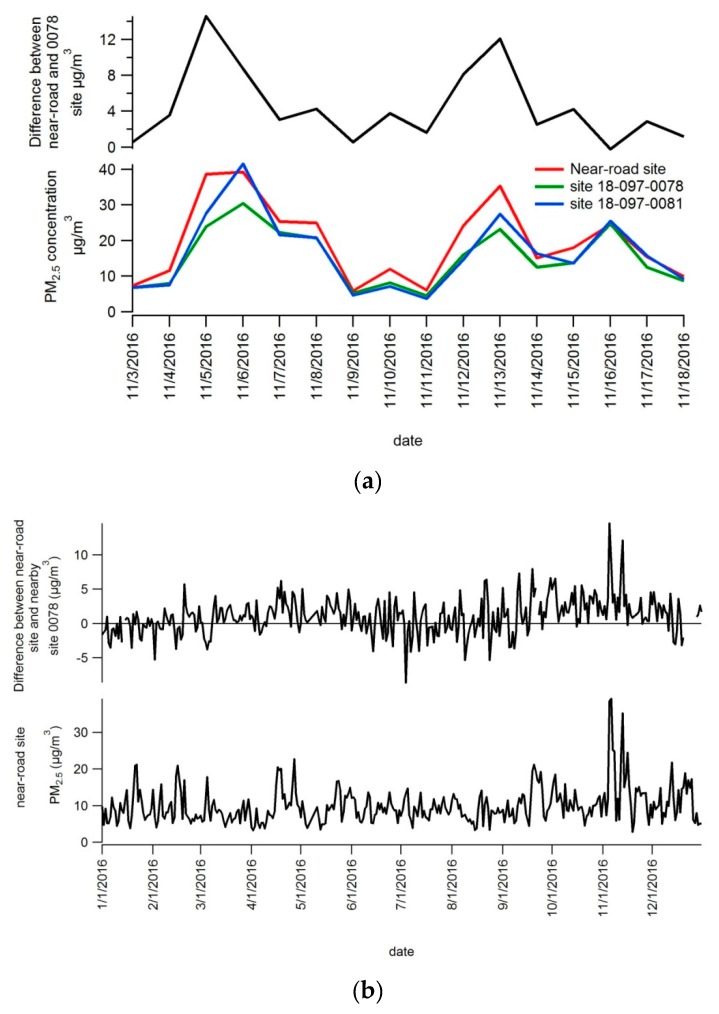
Indianapolis near-road PM_2.5_ increment. (**a**) 24-h PM_2.5_ concentrations at the Indianapolis-0087 near-road monitoring site and nearby sites (denoted by AQS ID) in Indianapolis from 3–18 November, 2016 (Washington Park, the background site, is 18-097-0078). Differences in 24-h PM_2.5_ concentrations between the near-road site and Washington Park (0078) are shown in the top panel. (**b**) 24-h PM_2.5_ concentration at Indianapolis-0087 and nearby Washington Park 0078 in 2016. The difference between these measurements is shown in the top panel.

**Figure 9 ijerph-16-01634-f009:**
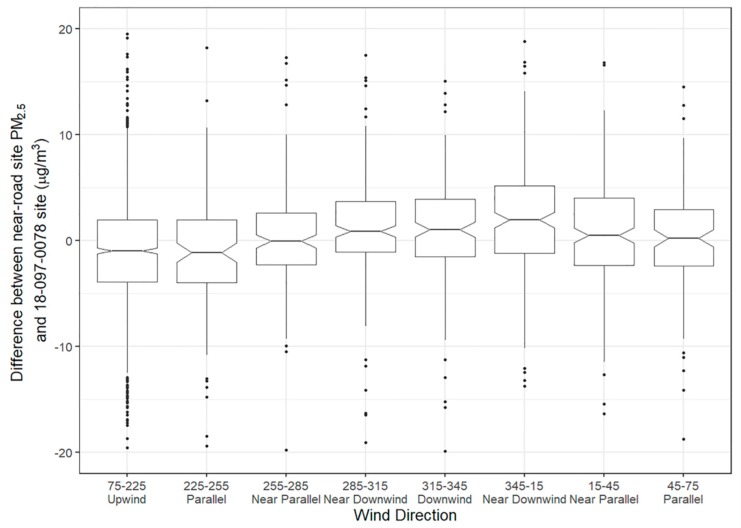
Notched box plot of Indianapolis hourly PM_2.5_ increment (difference between near-road and Washington Park site) by wind direction bin during 06:00 to 20:00 and when winds speeds are below 2 m/s. The notch shows the 95% confidence interval on the median and the boxes show the interquartile range (IQR); points beyond the IQR are within the notch or are plotted individually.

**Table 1 ijerph-16-01634-t001:** For the Denver near-road site, distances to and over roadway, median increment (all hours, 06:00–20:00 [“daytime”] with wind speed (ws) less than 2 m/s, and daytime hours with wind speed less than 1 m/s), and median wind speeds by wind direction bins.

Wind Direction	Wind Direction Bin Midpoint	Min Wind Direction	Max Wind Direction	Distance from Edge of Roadway (Far Shoulder) to Monitor (Meters)	Distance from Near Edge of Roadway to Monitor (Meters)	Distance over Road (Meters)	Median Increment, All Hours	Median ws, m/s	Median Increment (Daytime, ws < 2 m/s)	Median ws (Daytime, ws < 2 m/s)	Median Increment (Daytime, ws < 1 m/s)	Median ws (Daytime, ws < 1 m/s)
Upwind	90	15	165	-	-	-	1.6	1.7	1.9	1.34	2.05	0.72
Parallel	180	165	195	515	214	301	2.1	1.6	2.8	1.21	3.85	0.72
Near-parallel	210	195	225	127	40	87	2.5	1	2.95	0.98	3.5	0.76
Near-downwind	240	225	255	80	21	59	2.85	0.8	2.9	0.8	3.5	0.67
Downwind	270	255	285	66	16	50	1.7	1.2	2	1.07	2.3	0.58
Near-downwind	300	285	315	76	20	56	1.3	2.8	2	1.07	2.6	0.82
Near-parallel	330	315	345	98	36	62	1.3	1.9	1.9	1.21	2.1	0.67
Parallel	360	345	15	294	136	158	1.7	2.3	2.3	1.36	1.8	0.72

**Table 2 ijerph-16-01634-t002:** For Indianapolis near-road site, distances to and over roadway, median increment (all hours, 06:00–20:00 [“daytime”] with wind speed less than 2 m/s, and daytime hours with wind speed less than 1 m/s), and median wind speeds (ws) by wind direction bins.

Wind Direction	Wind Direction Bin Midpoint	Min Wind Direction	Max Wind Direction	Distance from Edge of Roadway (Far Shoulder) to Monitor (Meters)	Distance from Near Edge of Roadway to Monitor (Meters)	Distance over Road (Meters)	Median Increment	Median ws m/s	Median Increment (Daytime, ws < 2 m/s)	Median ws (Daytime, ws < 2 m/s)	Median Increment (Daytime, ws < 1 m/s)	Median ws (Daytime, ws < 1 m/s)
Upwind	150	75	225				-0.5	1.7	−0.98	1.2	−0.84	0.8
Parallel	240	225	255				-0.09	2.8	−1.16	1.5	−0.91	0.75
Near-parallel	270	255	285	164	62	102	1.31	2.4	−0.08	1.5	−0.21	0.8
Near-downwind	300	285	315	99	35	64	1.48	2.3	0.84	1.4	2.91	0.7
Downwind	330	315	345	83	30	53	1.37	1.9	1.02	1.3	0.47	0.8
Near-downwind	0	345	15	101	35	66	2.23	1.5	1.92	1.1	2.38	0.8
Near-parallel	30	15	45	189	65	124	2.05	1.6	0.47	1.5	0.67	0.8
Parallel	60	45	75				1.11	1.9	0.21	1.4	−0.57	0.8

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
