# Peer review of "Conditions Leading to Elevated PM2.5 at Near-Road Monitoring Sites: Case Studies in Denver and Indianapolis"

_ijerph, 2019, doi:10.3390/ijerph16091634_

Round 1
Reviewer 1 Report
In my mind, the article ijerph-456399 with the title "Conditions Leading to Elevated PM2.5 at Near-Road Monitoring Sites: Case Studies in Denver and Indianapolis" does not fulfil the quality standards of this journal. Although the near-road measurement of PM2.5 is important in urban air pollution monitoring, it is not clear in this manuscript what new information is being presented and how this research contributes to understanding of air pollution at roadsides. The paper is easy to read, but it shows no intriguing findings or only has some well-known results for the readers. Besides, the contents of the article are too simple to understand what important problems it can address. At the current state, I would not recommend it for publication. I encourage the authors to continue their work in order to reflect the unique value of this study.
Author Response
In my mind, the article ijerph-456399 with the title "Conditions Leading to Elevated PM2.5 at Near-Road Monitoring Sites: Case Studies in Denver and Indianapolis" does not fulfil the quality standards of this journal. Although the near-road measurement of PM2.5 is important in urban air pollution monitoring, it is not clear in this manuscript what new information is being presented and how this research contributes to understanding of air pollution at roadsides. The paper is easy to read, but it shows no intriguing findings or only has some well-known results for the readers. Besides, the contents of the article are too simple to understand what important problems it can address. At the current state, I would not recommend it for publication. I encourage the authors to continue their work in order to reflect the unique value of this study.
Thank you for the comments. Based on your feedback, and pairing it with recommendations from reviewer #2, we have added more quantitative analysis, including unique results looking at a year’s worth of hourly near-road PM2.5 data compared to nearby concentrations. We see a small but significant difference in PM2.5 concentrations under “true downwind” conditions compared to when winds are at an angle across the roadway, and compare these results to modeling and other observational studies. We believe that the combination of the case study on high PM2.5 days combined with an analysis of a year’s worth of hourly PM2.5 and meteorological data is unique, where we evaluate how the near-road “increment” can be higher under near-perpendicular conditions compared to conditions when the monitor is perpendicular to the roadway. We have provided additional text in the manuscript about the relevance and unique findings of the work.
Reviewer 2 Report
This paper presents findings from an analysis of near-road PM and NO2 concentrations with respect to traffic and meteorological conditions for two case studies, Denver and Indianapolis. These in-depth analyses complement existing national-scale studies examining similar relationships, and findings are consistent with previous near-road research. The paper is relevant in terms of researching the PM/NO2 relationship in greater detail. However, analyses are largely based on the visual interpretation of graphs. I believe that further analytical methods are needed to support the conclusions and for this reason recommend acceptance after major revisions.
Below are a list of both specific and general comments/recommendations:
Switch the pictures in Figure 1 so that Colorado is on the left and Indiana on the right to reflect the organization of the paper.
The authors mention correlations throughout the paper – are correlated, are not correlated, are poorly correlated – but there are no actual correlation statistics being made. Therefore, these words should not be used. For example, one should not reference this paper and state that “authors found correlations” because that would be misleading. I have some recommendations for performing correlations, but if no calculations are being made, then a different term should be used.
For correlations, please consider: 1) first, refer back to your DeWinter et al. (2018) reference to look for examples; 2) correlations between NO2 and PM; 3) correlations between concentrations and traffic counts; 4) possibly correlations between near-road and other monitors. This is the authors’ discretion but would help to support the findings; simple regressions are fine.
I am curious about the calculations, for example in Figure 3, of the difference between the near-road and the maximum of the other monitors. I am not familiar with this approach and am not sure it is representative. Please provide a reference or justification (for example, for not using the mean). Because site characteristics can vary widely, and be impacted by different sources and meteorological conditions, I am uncertain whether the difference between near-road and an arbitrary other monitor (whichever was the highest) for that given hour/day is more appropriate than the difference between near-road and an average of the other, regional, sites.
Another issue that should be addressed in more detail is the representation of downwind directions and their importance. Across the field, this is a challenge to represent given the unique topography and environmental conditions of a given site (e.g., search for “Venkatram roadway” references). The representation of upwind and downwind as full 180-degree arcs is misleading and affects interpretation of the wind roses. For example, for Colorado, a NNE wind direction (i.e., coming from the NNE) would provide a long fetch of roadway prior to impacting the site, and indeed, a large percentage of high concentrations come from this direction. Similarly for northward-blowing winds from the south. We would almost expect these to produce higher concentrations than a direct westerly wind that is traveling across very little roadway, even if it is perpendicularly downwind. This is consistent with modeling studies. In looking at Figures 3b and 4, you can almost see this pattern, where directions of 0-40, 160-200, and 260-300 are producing higher differences. These are almost suggestive of a direct near-road and wind direction influence. The same results are not necessarily found for Indianapolis, but perhaps this speaks to the micro-environmental differences between the two sites? I leave it to the authors to interpret based on this view and the suggested statistical analyses above.
I think the general conclusion that “PM concentrations were higher next to the road regardless of variations in daily traffic or meteorology” is basically supported but needs more statistical evidence. I also think that the Colorado wind rose has some interesting things going on that warrant scrutiny.
I commend the authors on the creative display of data. I think that their ability to display various forms of data on single graphs in a meaningful way is highly informative. This allows readers to examine the evidence in its entirety and conveys the information in a meaningful way.
The authors may also want to check the following references for some additional insights: Baldauf et al. (2012), JAWMA, Traffic and meteorological impacts on near-road air quality (https://doi.org/10.3155/1047-3289.58.7.865) and Hays et al. (2011) Atmos Env, Particle size distribution… (https://doi.org/10.1016/j.atmosenv.2010.11.010)
Author Response
This paper presents findings from an analysis of near-road PM and NO2 concentrations with respect to traffic and meteorological conditions for two case studies, Denver and Indianapolis. These in-depth analyses complement existing national-scale studies examining similar relationships, and findings are consistent with previous near-road research. The paper is relevant in terms of researching the PM/NO2 relationship in greater detail. However, analyses are largely based on the visual interpretation of graphs. I believe that further analytical methods are needed to support the conclusions and for this reason recommend acceptance after major revisions.
Below are a list of both specific and general comments/recommendations:
Switch the pictures in Figure 1 so that Colorado is on the left and Indiana on the right to reflect the organization of the paper.
done
The authors mention correlations throughout the paper – are correlated, are not correlated, are poorly correlated – but there are no actual correlation statistics being made. Therefore, these words should not be used. For example, one should not reference this paper and state that “authors found correlations” because that would be misleading. I have some recommendations for performing correlations, but if no calculations are being made, then a different term should be used.
For correlations, please consider: 1) first, refer back to your DeWinter et al. (2018) reference to look for examples; 2) correlations between NO2 and PM; 3) correlations between concentrations and traffic counts; 4) possibly correlations between near-road and other monitors. This is the authors’ discretion but would help to support the findings; simple regressions are fine.
We have added quantitative r2 values where we refer to correlations, and whether NO2 or NOx correlated with PM2.5. In addition we now highlight when differences in PM2.5 increment are statistically significant among different wind directions (perpendicular, near-perpendicular wind conditions, etc).
I am curious about the calculations, for example in Figure 3, of the difference between the near-road and the maximum of the other monitors. I am not familiar with this approach and am not sure it is representative. Please provide a reference or justification (for example, for not using the mean). Because site characteristics can vary widely, and be impacted by different sources and meteorological conditions, I am uncertain whether the difference between near-road and an arbitrary other monitor (whichever was the highest) for that given hour/day is more appropriate than the difference between near-road and an average of the other, regional, sites.
We have revised our calculations to show the difference between the near-road monitor and the nearest monitor, and oriented our discussion around this difference.
Another issue that should be addressed in more detail is the representation of downwind directions and their importance. Across the field, this is a challenge to represent given the unique topography and environmental conditions of a given site (e.g., search for “Venkatram roadway” references). The representation of upwind and downwind as full 180-degree arcs is misleading and affects interpretation of the wind roses. For example, for Colorado, a NNE wind direction (i.e., coming from the NNE) would provide a long fetch of roadway prior to impacting the site, and indeed, a large percentage of high concentrations come from this direction. Similarly for northward-blowing winds from the south. We would almost expect these to produce higher concentrations than a direct westerly wind that is traveling across very little roadway, even if it is perpendicularly downwind. This is consistent with modeling studies. In looking at Figures 3b and 4, you can almost see this pattern, where directions of 0-40, 160-200, and 260-300 are producing higher differences. These are almost suggestive of a direct near-road and wind direction influence. The same results are not necessarily found for Indianapolis, but perhaps this speaks to the micro-environmental differences between the two sites? I leave it to the authors to interpret based on this view and the suggested statistical analyses above.
We have conducted additional analyses that we report for both sites looking at hourly near-road increments by wind direction bin. As suggested in the Venkatram, Baryzk and other publications, we do see some significant (at 90% confidence) difference in concentrations under direct downwind conditions and conditions when the winds are at an angle across the roadway. Black carbon or ultrafine particles would likely yield a more significant signal, but such data are not available.
I think the general conclusion that “PM concentrations were higher next to the road regardless of variations in daily traffic or meteorology” is basically supported but needs more statistical evidence. I also think that the Colorado wind rose has some interesting things going on that warrant scrutiny.
We have now provided statistical interpretation of how PM2.5 varies with wind direction, traffic, etc. In particular, we statistically show that the PM2.5 increment is higher under near-perpendicular conditions versus perpendicular conditions.
I commend the authors on the creative display of data. I think that their ability to display various forms of data on single graphs in a meaningful way is highly informative. This allows readers to examine the evidence in its entirety and conveys the information in a meaningful way.
thank you
The authors may also want to check the following references for some additional insights: Baldauf et al. (2012), JAWMA, Traffic and meteorological impacts on near-road air quality (https://doi.org/10.3155/1047-3289.58.7.865) and Hays et al. (2011) Atmos Env, Particle size distribution… (https://doi.org/10.1016/j.atmosenv.2010.11.010)
Thank you for the suggestion. We have expanded the literature we cite, especially with regard to how concentrations vary by wind direction.
Round 2
Reviewer 1 Report
1. Please add more introduction about PM2.5 monitors used in both roadside and regional monitoring, which can make the readers clearly understand what measuring errors likely occur in the comparison among sites. The readers can also scientifically understand there may be just a measurement error between instruments rather than a real PM2.5 difference among sites in some days.2. The authors can clarify that they have got rid of the impacts of other meteorological factors, or those impacts can be neglected to an extent, because the author focused on discussing the impacts from wind and traffic volumes and speed in their study. As reported in previous studies (references as below), other dynamic factors such as relative humidity and air pressure can contribute to the variation of roadside PM2.5 concentrations. To make the research statement more rigorous and enrich the reference value of this study to future practice, I suggest the authors refer to some other studies on how multiple influencing factors influence the PM2.5 variation in road traffic environments.
I recommend this paper to be published at IJERPH after addressing the above two comments.
1. Richmond-Bryant, J., Snyder, M.G., Owen, R.C., et al. (2018). Factors associated with NO2 and NOx concentration gradients near a highway. Atmospheric Environment, 174, 214–226.
2. Richmond-Bryant, J., Saganich, C., Bukiewicz, L., et al. (2009). Associations of PM2.5 and black carbon concentrations with traffic, idling, background pollution, and meteorology during school dismissals. Science of the Total Environment, 407, 3357–3364.
3. Wang, Z., Zhong, S., He, H.D., et al. (2018). Fine-scale variations in PM2.5 and black carbon concentrations and corresponding influential factors at an urban road intersection. Building and Environment, 141, 215–225.
4. Wang, Z., Lu, Q.C., He, H.D., et al. (2017). Investigation of the spatiotemporal variation and influencing factors on fine particulate matter and carbon monoxide concentrations near a road intersection. Frontiers of Earth Science, 11(1), 63–75.
5. Pearce, J.L., Beringer, J., Nicholls, N., et al. (2011). Quantifying the influence of local meteorology on air quality using generalized additive models. Atmospheric Environment, 45, 1328–1336.
Author Response
Thank you for the comments.
Regarding comment 1:
Please add more introduction about PM2.5 monitors used in both roadside and regional monitoring, which can make the readers clearly understand what measuring errors likely occur in the comparison among sites. The readers can also scientifically understand there may be just a measurement error between instruments rather than a real PM2.5 difference among sites in some days.
We have added the specific monitors that are used at the near-road site and the nearby sites, as well as the comparison of the continuous instrument to the collocated FRM so that readers can evaluate how comparable the data from the hourly instruments are. Specifically, we added text on page 3, lines 121 and 135:
The PM2.5 was measured hourly via a GRIMM EDM 180 Federal Equivalent Method (FEM) at both the near-road and nearby CAMP site. The GRIMM at the near-road site has correlation (R) with the collocated 24-hour FRM measurement of 0.95 and slope of 0.91, for the 2015-2017 (see https://www.epa.gov/outdoor-air-quality-data/pm25-continuous-monitor-comparability-assessments).
The near-road site has hourly data via a Thermo Scientific 5030 SHARP instrument, and the nearby Washington Park site has hourly data via a MetOne BAM 1020, both of which are FEM. The daily averaged data at the near-road monitoring site has a correlation (R) of 0.93 and slope of 0.97 with the collocated FRM for 2015-2017, and the daily averaged data from the BAM at the Washington Park site has correlation (R) of 0.92 and slope of 0.89 with the collocated FRM.
Regarding comment 2:
The authors can clarify that they have got rid of the
impacts of other meteorological factors, or those impacts can be neglected to
an extent, because the author focused on discussing the impacts from wind and
traffic volumes and speed in their study. As reported in previous studies
(references as below), other dynamic factors such as relative humidity and air
pressure can contribute to the variation of roadside PM2.5 concentrations. To
make the research statement more rigorous and enrich the reference value of
this study to future practice, I suggest the authors refer to some other
studies on how multiple influencing factors influence the PM2.5 variation in
road traffic environments.
I recommend this paper to be published at
IJERPH after addressing the above two comments.
In the introduction, we have added discussion using the above and additional citations regarding the complex nature of PM2.5 next to roadways. We have now provided specific study examples where meteorological factors such as relative humidity, temperature and other measures were predictive of near-road pollution in GAMs or other models
Reviewer 2 Report
No need to go into detail. The paper is much improved. Thank you for making a sincere effort.
Author Response
Reviewer 2
No need to go into detail. The paper is much improved. Thank you for making a sincere effort.
Thank you for the recommendations to make this manuscript stronger.